# NETWORK EMBEDDING METHODS ARE STRONG BASELINES FOR LINK PREDICTION

## ABSTRACT

Due to their impressive performance across a wide range of graph-related tasks, graph neural networks (GNNs) have emerged as the dominant approach to link prediction, often assumed to outperform network embedding methods. However, their performance is hindered by the training–inference discrepancy and a strong reliance on high-quality node features. In this paper, we revisit classical network embedding methods within a unified training framework and highlight their conceptual continuity with the paradigms used in GNN-based link prediction. We further conduct an extensive empirical evaluation of three classical methods (LINE, DeepWalk, and node2vec) on standard link prediction benchmarks. Our findings suggest that the reported superiority of GNNs may be overstated, partly due to inconsistent training protocols and suboptimal hyperparameter choices for embedding-based methods. Notably, when incorporated into a modern link prediction framework with minimal configuration changes, these classical methods achieve state-of-the-art performance on both undirected and directed tasks. Despite being proposed nearly a decade ago, they outperform recent GNN models on 13 of 16 benchmark datasets. These results highlight the need for more rigorous and equitable evaluation practices in graph learning research.

## 1 INTRODUCTION

Network embedding provides a foundational framework for learning graph representations by mapping high-dimensional graph structures to low-dimensional vector spaces, while preserving topological relationships and node semantics (Cui et al., 2018). Due to their ability to model complex relational patterns, network embedding methods have been widely adopted in applications such as social network analysis and recommendation systems (Daud et al., 2020; Wen et al., 2018). Link prediction is a core task in graph mining that aims to infer missing or future connections and relies on accurate recognition of structural patterns. Traditional embedding methods achieve this task through inner product operations on learned representations, focusing exclusively on topology without explicit modeling of node attributes (Tang et al., 2015; Taskar et al., 2003). In contrast, Graph Neural Networks (GNNs) integrate both structural and attribute-based signals via message-passing mechanisms, garnering significant attention in recent years (Kipf & Welling, 2016; Kollias et al., 2022; Zhu et al., 2021; Yun et al., 2021; Wang et al., 2021; Tong et al., 2020b;a; Rossi et al., 2024). *This shift has positioned GNNs as the dominant approach in link prediction research, inadvertently marginalizing classical embedding approaches.*

Despite demonstrating superior performance, GNNs face inherent challenges in link prediction (Mao et al., 2024; Li et al., 2023). These include task-alignment issues, such as training–inference discrepancies, and the risk of data leakage (Zhu et al., 2024a; Wang et al., 2024; Zhang et al., 2021), which often necessitate techniques like edge masking to ensure valid evaluation (Wang et al., 2024). Moreover, GNN performance is highly sensitive to the quality of input node features (Zhu et al., 2024b). However, real-world applications of GNN frequently encounter two major limitations: (1) the absence of explicit node features in datasets such as anonymous social networks or biological interaction graphs (Boukharouba et al., 2023), and (2) sparse or noisy features that reduce their effectiveness. In such cases, traditional topology-driven embedding methods, such as DeepWalk (Perozzi et al., 2014) and LINE (Tang et al., 2015), are theoretically well-suited for the link prediction task. *Nevertheless, results reported in prior work often show that these embedding methods underperform compared to GNN models.* Additionally, GNNs typically require architectural modifications (e.g.,

direction-aware aggregation layers (Tong et al., 2020a; Kollias et al., 2022; Rossi et al., 2024)) to process directed graphs. In contrast, random walk-based embedding methods naturally incorporate edge direction through their sampling process, further highlighting their practicality.

These tensions motivate a critical re-examination of whether traditional network embedding methods have been undervalued in link prediction research. To address this question, we revisit three established methods (LINE (Tang et al., 2015), DeepWalk (Perozzi et al., 2014), and node2vec (Grover & Leskovec, 2016)) within a unified encoder-decoder framework, highlighting their alignment with modern GNN-based training paradigms. We re-implement and evaluate these methods under standardized training protocols across **16 real-world datasets** containing both **undirected and directed graphs**. Through comprehensive benchmarking in comparison with state-of-the-art GNNs under standardized evaluation protocols, our empirical analysis reveals these critical findings:

- **Classical network embedding methods, proposed nearly a decade ago, show highly competitive performance on both undirected and directed graphs**, outperforming state-of-the-art GNNs on 13 out of 16 datasets. These results not only reveal the underestimated potential of classical approaches in prior work but also establish their enduring viability as strong contenders for link prediction when implemented within modern training frameworks. Furthermore, **node features do not necessarily improve link prediction**, suggesting that the absence of node attributes in embedding-based methods is generally not a major limitation.
- Ablation studies reveal three critical principles: (1) **end-to-end training** significantly enhances the performance of embedding methods; (2) **expressive decoders**, particularly MLPs with Hadamard product input, often lead to better link prediction results; and (3) **pairwise ranking losses** generally result in better performance than pointwise losses on existing benchmarks.

## 2 PRELIMINARIES

Given a graph $G = (V, E)$, where $V$ is the set of nodes and $E$ is the set of edges. Let $n = |V|$ denote the number of nodes. Each edge $e = (v_i, v_j) \in E$ connects nodes $v_i$ and $v_j$, where $v_i$ and $v_j$ are neighbors. The neighborhood set of node $v_i$ is denoted by $N(v_i)$, and its degree is defined as $d_i = |N(v_i)|$. A directed edge $e = (v_i, v_j)$ denotes an asymmetric relationship from node $v_i$ to $v_j$. In contrast, undirected edges imply mutual connectivity. The adjacency matrix $\mathbf{A} \in \mathbb{R}^{n \times n}$ encodes connectivity, where $\mathbf{A}_{ij} = 1$ indicates the presence of an edge between nodes $v_i$ and $v_j$, and $\mathbf{A}_{ij} = 0$ otherwise. For undirected graphs, the adjacency matrix is symmetric, i.e., $\mathbf{A}_{ij} = \mathbf{A}_{ji}$.

**Network Embedding** aims to map the structural information of a graph into a low-dimensional vector space, assigning each node a compact representation that captures its structural role. These representations effectively preserve key graph properties and are widely applicable to various graph-based machine learning tasks, including node classification, node clustering, and link prediction (Cui et al., 2018; Yin & Wei, 2019). Formally, the goal is to learn a vector representation $\mathbf{e}_i \in \mathbb{R}^d$ for each node $v_i \in V$, where $d \ll n$ and $d$ denotes the embedding size.

**LINE** (Tang et al., 2015) is a network embedding method that supports various graph structures, including directed and undirected graphs. Its core objective is preserving first- and second-order proximities, enabling the model to capture local and global structural patterns. First-order proximity describes direct connections between nodes, while second-order proximity reflects structural similarity based on shared neighborhoods, allowing the model to identify relationships between nodes that are not directly connected. To model these relationships, LINE learns two embedding vectors for each node $v_i \in V$: a content embedding $\mathbf{e}_i$ and a context embedding $\mathbf{e}_i'$. To address the computational cost of evaluating all node pairs, LINE uses negative sampling (Mikolov et al., 2013b) and the Alias Method (Li et al., 2014) for efficient sampling. The model optimizes an objective function based on observed and negatively sampled node pairs:

$$L = \sum_{i=1}^{|V|} \sum_{j=1}^{|V|} \mathbf{A}_{i,j} \log \sigma \left( \mathbf{e}_i^\top \mathbf{e}_j' \right) + b \sum_{i=1}^{|V|} d_i \mathbb{E}_{j' \sim P_N} \left[ \log \sigma \left( -\mathbf{e}_i^\top \mathbf{e}_{j'}' \right) \right], \quad (1)$$

where $P_N$ denotes the negative sampling distribution, $b$ is the number of negative sampling, and $d_i$ is the degree of node $v_i$. In the case of directed graphs, $d_i$ corresponds to the out-degree.

**DeepWalk** (Perozzi et al., 2014) is a network embedding method inspired by the word2vec (Mikolov et al., 2013a) model, which is widely used in natural language processing. DeepWalk draws an

analogy between graph structures and language models by treating nodes as words and random walk sequences as sentences. Specifically, for each node $v \in V$, DeepWalk performs multiple random walks of length $K$ to generate node sequences:

$$\text{RW}(v) = \{v_0 = v, v_1, \ldots, v_K\}, \quad v_{i+1} \sim \mathcal{U}\left(N(v_i)\right), \tag{2}$$

where $v_0 = v$ denotes the random walk starting from node $v$, and $\mathcal{U}$ denotes the uniform distribution over the neighbors $N(v_i)$. These sequences form the training corpus for the Skip-Gram model (Levy & Goldberg, 2014), which learns node embeddings by maximizing the probability of predicting context nodes within a window of size $K$ around each target node.

**node2vec** (Grover & Leskovec, 2016) introduces a biased random walk that enables a flexible trade-off between breadth-first search (BFS) and depth-first search (DFS). Instead of sampling uniformly from the neighbors of the current node, node2vec uses information from the previous node in the walk to compute transition probabilities. Given a step $(t, v)$, the transition probability to each neighbor $x$ of the current node $v$ is adjusted by two parameters, $p$ and $q$, which control the walk's behavior. Specifically, $p$ governs the likelihood of returning to $t$, while $q$ affects the tendency to explore further nodes. These adjustments depend on the shortest path distance $d_{tx}$ between nodes $t$ and $x$. The transition probability $\pi_{vx} = \alpha_{pq}(t, x)$ is defined as:

$$\alpha_{pq}(t, x) = \begin{cases} \frac{1}{p}, & \text{if } d_{tx} = 0 \\ 1, & \text{if } d_{tx} = 1 \\ \frac{1}{q}, & \text{if } d_{tx} = 2 \end{cases}. \tag{3}$$

When $p = q = 1$, node2vec reduces to DeepWalk, resulting in uniform sampling from the neighbors.

**Link Prediction** aims to predict the likelihood of unobserved or future edges between node pairs in a graph. For a node pair $(v_i, v_j)$, the corresponding embeddings $\mathbf{e}_i$ and $\mathbf{e}_j$ are passed through a decoding function $g(\cdot)$ (e.g., dot product or MLP) to compute a prediction score: $\hat{y}_{ij} = g(\mathbf{e}_i, \mathbf{e}_j)$, where $\hat{y}_{ij} \in [0, 1]$ represents the estimated probability of an edge existing between $v_i$ an $v_j$. The training objective minimizes the total loss over the training edge: $L = \sum_{(u,v) \in E_{\text{train}}} \ell\left(\hat{y}_{uv}, y_{uv}\right)$, where $y_{uv} \in \{0, 1\}$ denotes the ground-truth label (1 for existing edges, 0 otherwise). The training set $E_{\text{train}}$ includes observed positive edges and sampled negative edges from non-connected node pairs. The loss function $\ell(\hat{y}_{uv}, y_{uv})$ measures the discrepancy between the predicted score $\hat{y}_{uv}$ and the ground-truth label $y_{uv}$, and is typically implemented using binary cross-entropy (Tang et al., 2015) or a ranking-based loss (Zhang et al., 2024; Rosenfeld et al., 2014).

**Undirected and Directed Graphs**. Link prediction can be performed on both undirected and directed graphs. Classical embedding methods such as LINE (Tang et al., 2015), DeepWalk (Perozzi et al., 2014) and node2vec (Grover & Leskovec, 2016) are compatible with both types. In particular, random walk–based methods (e.g., DeepWalk and node2vec) inherently support directionality through the design of the walk process, while LINE directly supports edge-level interactions.

**Unified View: An Encoder-Decoder Perspective**. LINE, DeepWalk, and node2vec can be formally interpreted within the modern representation learning paradigm as instances of an encoder–decoder framework. Positive samples generated from graph structure or random walks serve as a form of data augmentation. The process of learning node embeddings acts as the encoder, while a decoder, typically defined using inner products or similarity functions, estimates the likelihood of edge formation between node pairs. The objective function then guides learning by measuring the discrepancy between predicted and true edge labels. This unified perspective provides a conceptual foundation for analyzing training strategies across embedding models, which we will explore in the next section.

## 3  A UNIFIED ENCODER-DECODER FRAMEWORK FOR LINK PREDICTION

In this section, we describe a widely adopted framework for training link prediction models, consisting of four essential components: data augmentation, encoder, decoder, and loss function. Each component is detailed below. This encoder–decoder paradigm has become standard in recent link prediction research (Zhang et al., 2024; Wang et al., 2021; He et al., 2025). Popular GNNs such as GCN (Kipf & Welling, 2016), GAT (Veličković et al., 2018) and HL-GNN (Zhang et al., 2024) are typically used as encoders, while the other components (data augmentation strategy, decoder function, and loss objective) are chosen independently as hyperparameters. Importantly, we argue that

classical embedding methods such as LINE (Tang et al., 2015), DeepWalk (Perozzi et al., 2014) and node2vec (Grover & Leskovec, 2016) can be naturally adapted to this framework. By mapping their components, such as sampling strategies and embedding tables, to appropriate choices of data augmentation and encoder design, these methods can be trained under the same protocol as GNN-based models. This alignment enables fair comparisons and underscores the continuity between classical and modern link prediction approaches.

**Data Augmentation**. Incorporating high-order structural information is essential for improving link prediction performance. Although deeper GNN architectures can theoretically capture higher-order neighborhoods, they often suffer from over-smoothing and increased computational complexity (Keriven, 2022; Peng et al., 2024). As a more efficient alternative, random walk–based augmentation introduces high-order context at the input level and has been adopted in several recent studies (Wang et al., 2021; Zhang et al., 2024). Specifically, for a given node $v$, a random walk of length $K$ generates a sequence $\mathrm{RW}(v) = \{v_0 = v, v_1, \ldots, v_K\}$. The original edge set $E$ is then augmented by adding additional positive pairs based on node co-occurrence in the walk sequences:

$$E_{\mathrm{aug}} = E \cup \{(v_i, v_j) \mid v_j \in \mathrm{RW}(v_i), \forall v_i \in V\}. \tag{4}$$

This augmentation strategy is conceptually consistent with the sampling procedures used in classical network embedding methods such as DeepWalk (Perozzi et al., 2014) and node2vec (Grover & Leskovec, 2016), where random walks define context windows and positive training pairs. Therefore, random walk–based data augmentation can be viewed as a direct extension or reinterpretation of these traditional techniques, reinforcing the continuity between classical and modern approaches in graph representation learning.

**Encoder** is responsible for generating low-dimensional representations for each node in the graph, based on its structural context and, if available, node features. Formally, given a graph $G = (V, E)$ and an optional node feature matrix $\mathbf{X} \in \mathbb{R}^{n \times h}$, where $h$ is the number of features, the encoder maps the inputs to a node embedding matrix $\mathbf{E} \in \mathbb{R}^{n \times d}$: $\mathbf{E} = f(G, \mathbf{X})$.

- In GNN-based models (Kipf & Welling, 2016; Xu et al., 2020; Zhang et al., 2024), the encoder function $f(\cdot)$ is typically implemented as a message-passing neural network that takes both the adjacency matrix $\mathbf{A}$ and the node feature matrix $\mathbf{X}$ as input: $\mathbf{E} = \mathrm{GNN}(\mathbf{A}, \mathbf{X}; \theta)$, where $\theta$ denotes the model parameters. GNNs integrate node features and graph structure in an end-to-end manner, making them particularly effective when rich feature information is available.
- In classical network embedding methods such as LINE (Tang et al., 2015), the embedding matrix $\mathbf{E} \in \mathbb{R}^{n \times d}$ itself constitutes the model parameters. These methods are typically feature-agnostic, relying solely on the graph structure. When node features are available, the learned embeddings can optionally be concatenated with $\mathbf{X}$ to form a hybrid representation: $\overline{\mathbf{E}} = [\mathbf{E} \| \mathbf{X}]$. This combined representation can then be used for prediction tasks or as input to downstream models.

**Decoder** computes a similarity score between two node embeddings, reflecting the likelihood of an edge. Formally, given embeddings $\mathbf{e}_i$ and $\mathbf{e}_j$, the decoder outputs a score $\hat{y}_{ij} = g(\mathbf{e}_i, \mathbf{e}_j)$. A widely used choice is the dot product decoder, defined as $\hat{y}_{ij} = \mathbf{e}_i^\top \mathbf{e}_j$, which is computationally efficient and commonly adopted in classical methods (Tang et al., 2015; Perozzi et al., 2014; Yin & Wei, 2019; Grover & Leskovec, 2016). To better capture the relational patterns between node pairs, more expressive decoders can be employed. A typical alternative applies a multi-layer perceptron (MLP) to a combined representation of the two embeddings. Formally, the predicted score is computed as $\hat{y}_{ij} = \mathrm{MLP}(\mathbf{e}_i \circ \mathbf{e}_j)$, where $\circ$ denotes a composition operation such as element-wise (Hadamard) product or concatenation. MLP-based decoders offer greater modeling flexibility and enable the use of standard deep learning techniques such as dropout and non-linear activations, which can enhance generalization and improve the quality of learned embeddings.

**Loss Function**. Commonly used loss functions in link prediction can be broadly categorized into two classes: pointwise losses and pairwise ranking losses.

- **Binary Cross-Entropy (BCE)** is a standard pointwise loss function that models link prediction as a binary classification problem. The BCE loss is defined over the training edge set as:

$$L_{\mathrm{BCE}} = - \sum_{(v_i, v_j) \in E_{\mathrm{train}}} \left[ y_{ij} \log \sigma(\hat{y}_{ij}) + (1 - y_{ij}) \log(1 - \sigma(\hat{y}_{ij})) \right], \tag{5}$$

where $\hat{y}_{ij}$ is the decoder's output score for the node pair $v_i$ and $v_j$, and $\sigma(\cdot)$ denotes the sigmoid function, which maps the score to a probability.

Table 1: Statistics of datasets used in the experiments.

| Dataset | #Nodes | #Edges | #Features | Avg. Degree | Direction | Domain |
|---|---|---|---|---|---|---|
| Cora | 2,708 | 5,278 | 1,433 | 3.90 | ✘ | citation network |
| CiteSeer | 3,327 | 4,676 | 3,703 | 2.81 | ✘ | citation network |
| Pubmed | 18,717 | 44,327 | 500 | 4.73 | ✘ | citation network |
| Photo | 7,650 | 238,162 | 745 | 62.26 | ✘ | social network |
| Computers | 13,752 | 491,722 | 767 | 71.51 | ✘ | co-purchase network |
| ogbl-collab | 235,868 | 1,285,465 | 128 | 5.45 | ✘ | collaboration network |
| ogbl-ddi | 4,267 | 1,334,889 | - | 312.84 | ✘ | drug-drug interaction |
| ogbl-ppa | 576,289 | 30,326,273 | 58 | 52.62 | ✘ | protein-protein association |
| ogbl-citation2 | 2,927,963 | 30,561,187 | 128 | 10.44 | ✘ | citation network |
| Cora-ML | 2,810 | 8,229 | 2,879 | 5.9 | ✔ | citation network |
| CiteSeer-D | 2,110 | 3,705 | 3,703 | 3.5 | ✔ | citation network |
| Photo-D | 7,487 | 143,590 | 745 | 38.4 | ✔ | co-purchasing network |
| Computers-D | 13,381 | 287,076 | 767 | 42.9 | ✔ | co-purchasing network |
| WikiCS | 11,311 | 290,447 | 300 | 51.3 | ✔ | weblink network |
| Slashdot | 74,444 | 424,557 | - | 11.4 | ✔ | social network |
| Epinions | 100,751 | 708,715 | - | 14.1 | ✔ | social network |

- **Bayesian Personalized Ranking (BPR)** is a widely used pairwise ranking loss, originally proposed for modeling implicit feedback in recommender systems (Rendle et al., 2012). Its core idea is to optimize the ranking between positive and negative samples, ensuring that the predicted score of a positive sample is higher than that of a negative one. Instead of working on individual pairs in $E_{\text{train}}$, BPR operates on triplets $\mathcal{O}$, where each triplet $(v_i, v_j, v_k)$ consists of a target node $v_i$, a positive node $v_j$ and a negatively sampled node $v_k$. The BPR loss is formulated as:

$$L_{\text{BPR}} = - \sum_{(v_i,v_j,v_k)\in\mathcal{O}} \log \sigma \left( \hat{y}_{ij} - \hat{y}_{ik} \right), \tag{6}$$

where $\hat{y}_{ij}$ and $\hat{y}_{ik}$ denote the scores for the positive and negative node pairs, respectively.

While BCE treats each edge independently as a binary classification task, BPR focuses on the relative ranking of positive and negative pairs, making it well-suited for top-$k$ link prediction. Its objective also aligns naturally with ranking-based metrics such as Hit Rate and MRR (He et al., 2017; Rendle et al., 2012). We also adopt the AUC loss (Zhang et al., 2024; Rosenfeld et al., 2014), a pairwise ranking loss that encourages higher scores for positive links than negative. This loss is widely used (Zhang et al., 2024; Wang et al., 2021) and included in our experiments.

## 4 EXPERIMENTAL SETUP FOR LINK PREDICTION

**Datasets**. Our empirical evaluation covers 16 real-world graph datasets, including both undirected and directed graphs, with key statistics summarized in Table 1.

- **Undirected Graphs**. We use nine widely adopted datasets for evaluating undirected link prediction. Specifically, (1) Cora, (2) CiteSeer, and (3) Pubmed are sourced from the Planetoid dataset collection (Sen et al., 2008); (4) Photo and (5) Computers are taken from the Amazon co-purchase networks (Shchur et al., 2018); (6) ogbl-collab, (7) ogbl-ddi, (8) ogbl-ppa, and (9) ogbl-citation2 are provided by the Open Graph Benchmark (OGB) (Hu et al., 2020). For all these datasets, we follow the data processing, edge splitting protocols, and evaluation metrics as defined in prior work (Chamberlain et al., 2023; Zhang et al., 2024), to ensure fairness and comparability.
- **Directed Graphs**. We also conduct experiments on seven directed link prediction datasets provided by DirLinkBench (He et al., 2025). These include two citation networks: (1) Cora-ML and (2) CiteSeer-D; two co-purchasing networks: (3) Photo-D and (4) Computers-D; one web link network: (5) WikiCS; and two social networks: (6) Slashdot and (7) Epinions. For fairness and consistency, we adopt the provided data splits, formats, and task definitions.

**Baseline**. We take three classical network embedding methods (LINE, DeepWalk, and node2vec) as representative examples and compare their performance with GNN-based models on both undirected and directed link prediction tasks:

- **For undirected link prediction**, our evaluation covers a comprehensive set of baselines. These comprise traditional heuristic approaches such as CN (Barabási & Albert, 1999), RA (Zhou et al.,

Table 2: Comparison of link prediction performance on undirected graphs. OOM indicates methods that exceeded memory limits. Models marked with * denote our implementations. Results ranked **first**, **second**, and **third** are highlighted.

| Method | Cora Hits@100 | CiteSeer Hits@100 | Pubmed Hits@100 | Photo AUC | Computers AUC | collab Hits@50 | ddi Hits@20 | ppa Hits@100 | citation2 MRR |
|--------|------|------|------|------|------|------|------|------|------|
| CN | $33.92_{\pm0.46}$ | $29.79_{\pm0.90}$ | $23.13_{\pm0.15}$ | $96.73_{\pm0.00}$ | $96.15_{\pm0.00}$ | $56.44_{\pm0.00}$ | $17.73_{\pm0.00}$ | $27.65_{\pm0.00}$ | $51.47_{\pm0.00}$ |
| RA | $41.07_{\pm0.48}$ | $33.56_{\pm0.17}$ | $27.03_{\pm0.35}$ | $97.20_{\pm0.00}$ | $96.82_{\pm0.00}$ | $64.00_{\pm0.00}$ | $27.60_{\pm0.00}$ | $49.33_{\pm0.00}$ | $51.98_{\pm0.00}$ |
| KI | $42.34_{\pm0.39}$ | $35.62_{\pm0.33}$ | $30.91_{\pm0.69}$ | $97.45_{\pm0.00}$ | $97.05_{\pm0.00}$ | $59.79_{\pm0.00}$ | $21.23_{\pm0.00}$ | $24.31_{\pm0.00}$ | $47.83_{\pm0.00}$ |
| RWR | $42.57_{\pm0.56}$ | $36.78_{\pm0.58}$ | $29.77_{\pm0.45}$ | $97.51_{\pm0.00}$ | $96.98_{\pm0.00}$ | $60.06_{\pm0.00}$ | $22.01_{\pm0.00}$ | $22.16_{\pm0.00}$ | $45.76_{\pm0.00}$ |
| GCN | $66.79_{\pm1.65}$ | $67.08_{\pm2.94}$ | $53.02_{\pm1.39}$ | $98.61_{\pm0.15}$ | $98.55_{\pm0.27}$ | $47.14_{\pm1.45}$ | $37.07_{\pm5.07}$ | $18.67_{\pm1.32}$ | $84.74_{\pm0.21}$ |
| GAT | $60.78_{\pm3.17}$ | $62.94_{\pm2.45}$ | $46.29_{\pm1.73}$ | $98.42_{\pm0.19}$ | $98.47_{\pm0.32}$ | $55.78_{\pm1.39}$ | $54.12_{\pm5.43}$ | $19.94_{\pm1.69}$ | $86.33_{\pm0.54}$ |
| SEAL | $81.71_{\pm1.30}$ | $83.89_{\pm2.15}$ | $75.54_{\pm1.32}$ | $98.85_{\pm0.04}$ | $98.70_{\pm0.18}$ | $64.74_{\pm0.43}$ | $30.56_{\pm3.86}$ | $48.80_{\pm3.16}$ | $87.67_{\pm0.32}$ |
| NBFNet | $71.65_{\pm2.27}$ | $74.07_{\pm1.75}$ | $58.73_{\pm1.99}$ | $98.29_{\pm0.35}$ | $98.03_{\pm0.54}$ | OOM | $4.00_{\pm0.58}$ | OOM | OOM |
| Neo-GNN | $80.42_{\pm1.31}$ | $84.67_{\pm2.16}$ | $73.93_{\pm1.19}$ | $98.74_{\pm0.55}$ | $98.27_{\pm0.79}$ | $62.13_{\pm0.58}$ | $63.57_{\pm3.52}$ | $49.13_{\pm0.60}$ | $87.26_{\pm0.84}$ |
| BUDDY | $88.00_{\pm0.44}$ | $92.93_{\pm0.27}$ | $74.10_{\pm0.78}$ | $99.05_{\pm0.21}$ | $98.69_{\pm0.34}$ | $65.94_{\pm0.58}$ | $78.51_{\pm1.36}$ | $49.85_{\pm0.20}$ | $87.56_{\pm0.11}$ |
| HL-GNN | $94.22_{\pm1.64}$ | $94.31_{\pm1.51}$ | $88.15_{\pm0.38}$ | $99.11_{\pm0.07}$ | $98.82_{\pm0.21}$ | $68.11_{\pm0.54}$ | $80.27_{\pm3.98}$ | $56.77_{\pm0.84}$ | $89.43_{\pm0.83}$ |
| MF | $64.67_{\pm1.43}$ | $65.19_{\pm1.47}$ | $46.94_{\pm1.27}$ | $97.92_{\pm0.37}$ | $97.56_{\pm0.66}$ | $38.86_{\pm0.29}$ | $13.68_{\pm4.75}$ | $32.29_{\pm0.94}$ | $51.86_{\pm4.43}$ |
| DeepWalk | $70.34_{\pm2.96}$ | $72.05_{\pm2.56}$ | $54.91_{\pm1.25}$ | $98.83_{\pm0.23}$ | $98.45_{\pm0.45}$ | $50.37_{\pm0.34}$ | $26.42_{\pm6.10}$ | $35.12_{\pm0.79}$ | $55.58_{\pm1.75}$ |
| node2vec | $68.43_{\pm2.65}$ | $69.34_{\pm3.04}$ | $51.88_{\pm1.55}$ | $98.37_{\pm0.33}$ | $98.21_{\pm0.39}$ | $48.88_{\pm0.54}$ | $23.26_{\pm2.09}$ | $22.26_{\pm0.88}$ | $61.41_{\pm0.11}$ |
| **LINE*** | $91.63_{\pm1.02}$ | $95.71_{\pm0.94}$ | $81.08_{\pm0.31}$ | $99.10_{\pm0.01}$ | $98.97_{\pm0.01}$ | $67.89_{\pm0.70}$ | $90.13_{\pm3.04}$ | $67.49_{\pm1.35}$ | $89.77_{\pm1.10}$ |
| **DeepWalk*** | $94.36_{\pm1.59}$ | $95.25_{\pm1.91}$ | $87.36_{\pm0.52}$ | $99.10_{\pm0.02}$ | $98.82_{\pm0.01}$ | $68.79_{\pm0.51}$ | $79.01_{\pm1.27}$ | $58.36_{\pm1.51}$ | $81.05_{\pm1.48}$ |
| **node2vec*** | $94.50_{\pm0.81}$ | $95.89_{\pm1.32}$ | $88.04_{\pm0.42}$ | $99.12_{\pm0.01}$ | $98.83_{\pm0.02}$ | $68.92_{\pm0.55}$ | $79.14_{\pm1.29}$ | $59.28_{\pm1.34}$ | $81.68_{\pm1.31}$ |

2009), KI (Katz, 1953), and RWR (Brin & Page, 1998); common GNN architectures including GCN (Kipf & Welling, 2016) and GAT (Veličković et al., 2018); and state-of-the-art models such as SEAL (Zhang & Chen, 2018), NBFNet (Zhu et al., 2021), Neo-GNN (Yun et al., 2021), BUDDY (Chamberlain et al., 2023), and HL-GNN (Zhang et al., 2024). The baselines also include embedding-based methods, such as MF (Koren et al., 2009), node2vec (Grover & Leskovec, 2016), and DeepWalk (Lian et al., 2018), widely used in earlier work. However, their reported performance is often much lower than that of GNN models, reinforcing the belief that they are no longer competitive. All baseline results reported are sourced from HL-GNN.

- **For directed link prediction**, baselines include classical approaches such as MLP, GCN (Kipf & Welling, 2016), GAT (Veličković et al., 2018), and APPNP (Gasteiger et al., 2019), as well as state-of-the-art methods for directed graphs, including DGCN (Tong et al., 2020b), DiGCN (Tong et al., 2020a), DiGCNIB (Tong et al., 2020a), DirGNN (Rossi et al., 2024), DHYPR (Zhou et al., 2022), DiGAE (Kollias et al., 2022), and SDGAE (He et al., 2025). Embedding-based methods such as STRAP (Yin & Wei, 2019), ODIN (Yoo et al., 2023), and ELTRA (Rehyani Hamedani et al., 2023) also show strong performance on several DirLinkBench datasets, outperforming some GNNs designed for directed graphs. All baseline results reported in this study are sourced from DirLinkBench, which provides a unified implementation and evaluation protocol.

**Implemental Setting**. To limit experimental workload and ensure fairness in comparison, we strictly follow the implementation settings, data processing procedures, and evaluation protocols defined in HL-GNN (Zhang et al., 2024) (for undirected graphs) and DirLinkBench (He et al., 2025) (for directed graphs). For each model, we report the mean performance and standard deviation over 10 runs with different random initializations. Models marked with * denote our re-implementations of classical methods under the unified framework described in Section 3. Specifically, LINE (Tang et al., 2015), DeepWalk (Perozzi et al., 2014) and node2vec (Grover & Leskovec, 2016) serve as encoders, while random walk sampling strategies from DeepWalk and node2vec are employed as data augmentation. The decoder is selected from {DOT, MLP (concat), MLP (Hadamard product)}, and the loss function is chosen from {BCE, BPR, AUC}, depending on the training configuration. Full implementation details and hyperparameter configurations can be found in Appendix D.

## 5 EXPERIMENTAL RESULTS AND FINDINGS

### 5.1 PERFORMANCE OF NETWORK EMBEDDING METHODS IN LINK PREDICTION

We present a detailed analysis of the performance comparison between network embedding methods and state-of-the-art GNNs on link prediction. As shown in Table 2 and Table 3, results across 16 datasets indicate that embedding methods often outperform or closely match advanced GNNs. Notably, they rank in the top two on all datasets and achieve the best performance on 13 datasets, demonstrating strong competitiveness. Several key observations are outlined below.

Table 3: Comparison of link prediction performance on directed graphs under the Hits@100 metric. OOM and TO indicate methods that exceeded memory limits and did not complete within 24 hours, respectively. Models marked with * denote our implementations. Results ranked **first**, **second**, and **third** are highlighted.

| Method | Cora-ML | CiteSeer-D | Photo-D | Computers-D | WikiCS | Slashdot | Epinions |
|---|---|---|---|---|---|---|---|
| MLP | $60.61_{\pm6.64}$ | $70.27_{\pm3.40}$ | $20.91_{\pm4.18}$ | $17.57_{\pm0.85}$ | $12.99_{\pm0.68}$ | $32.97_{\pm0.51}$ | $44.59_{\pm1.62}$ |
| GCN | $70.15_{\pm3.01}$ | $80.36_{\pm3.07}$ | $58.77_{\pm2.96}$ | $43.77_{\pm1.75}$ | $38.37_{\pm1.51}$ | $33.16_{\pm1.22}$ | $46.10_{\pm1.37}$ |
| GAT | $79.72_{\pm3.07}$ | $85.88_{\pm4.98}$ | $58.06_{\pm4.03}$ | $40.74_{\pm3.22}$ | $40.47_{\pm4.10}$ | $30.16_{\pm3.11}$ | $43.65_{\pm4.88}$ |
| APPNP | $86.02_{\pm2.88}$ | $83.57_{\pm4.90}$ | $47.51_{\pm2.51}$ | $32.24_{\pm1.40}$ | $20.23_{\pm1.72}$ | $33.76_{\pm1.05}$ | $41.99_{\pm1.23}$ |
| DGCN | $63.32_{\pm2.59}$ | $68.97_{\pm3.39}$ | $51.61_{\pm6.33}$ | $39.92_{\pm1.94}$ | $25.91_{\pm4.10}$ | TO | TO |
| DiGCN | $63.21_{\pm5.72}$ | $70.95_{\pm4.67}$ | $40.17_{\pm2.38}$ | $27.51_{\pm1.67}$ | $25.31_{\pm1.84}$ | TO | TO |
| DiGCNIB | $80.57_{\pm3.21}$ | $85.32_{\pm3.70}$ | $48.26_{\pm3.98}$ | $32.44_{\pm1.85}$ | $28.28_{\pm2.44}$ | TO | TO |
| DirGNN | $76.13_{\pm2.85}$ | $76.83_{\pm4.24}$ | $49.15_{\pm3.62}$ | $35.65_{\pm1.30}$ | $50.48_{\pm0.85}$ | $41.74_{\pm1.15}$ | $50.10_{\pm2.06}$ |
| MagNet | $56.54_{\pm2.95}$ | $65.32_{\pm3.26}$ | $13.89_{\pm0.32}$ | $12.85_{\pm0.59}$ | $10.81_{\pm0.46}$ | $31.98_{\pm1.06}$ | $28.01_{\pm1.72}$ |
| DUPLEX | $69.00_{\pm2.52}$ | $73.39_{\pm3.42}$ | $17.94_{\pm0.66}$ | $17.90_{\pm0.71}$ | $8.52_{\pm0.60}$ | $18.42_{\pm2.59}$ | $16.50_{\pm4.34}$ |
| DHYPR | $86.81_{\pm1.60}$ | $92.32_{\pm3.72}$ | $20.93_{\pm2.41}$ | TO | TO | OOM/TO | OOM/TO |
| DiGAE | $82.06_{\pm2.51}$ | $83.64_{\pm3.21}$ | $55.05_{\pm2.36}$ | $41.55_{\pm1.62}$ | $29.21_{\pm1.36}$ | $41.95_{\pm0.93}$ | $55.14_{\pm1.96}$ |
| SDGAE | $90.37_{\pm1.33}$ | $93.69_{\pm3.68}$ | $68.84_{\pm2.35}$ | $53.79_{\pm1.56}$ | $54.67_{\pm2.50}$ | $42.42_{\pm1.15}$ | $55.91_{\pm1.77}$ |
| STRAP | $79.09_{\pm1.57}$ | $69.32_{\pm1.29}$ | $69.16_{\pm1.44}$ | $51.87_{\pm2.07}$ | $76.27_{\pm0.92}$ | $31.43_{\pm1.21}$ | $58.99_{\pm0.82}$ |
| ODIN | $54.85_{\pm2.53}$ | $63.95_{\pm2.98}$ | $14.13_{\pm1.92}$ | $12.98_{\pm1.47}$ | $9.83_{\pm0.47}$ | $34.17_{\pm1.19}$ | $36.91_{\pm0.47}$ |
| ELTRA | $87.45_{\pm1.48}$ | $84.97_{\pm1.90}$ | $20.63_{\pm1.93}$ | $14.74_{\pm1.55}$ | $9.88_{\pm0.70}$ | $33.44_{\pm1.00}$ | $41.63_{\pm2.53}$ |
| **LINE*** | $88.15_{\pm0.80}$ | $86.13_{\pm1.32}$ | $67.28_{\pm2.59}$ | $51.03_{\pm3.76}$ | $72.33_{\pm3.39}$ | $42.24_{\pm0.64}$ | $59.89_{\pm1.59}$ |
| **DeepWalk*** | $92.42_{\pm1.31}$ | $92.54_{\pm1.35}$ | $66.98_{\pm1.55}$ | $51.01_{\pm1.29}$ | $75.41_{\pm1.25}$ | $38.65_{\pm0.97}$ | $59.56_{\pm1.48}$ |
| **node2vec*** | $92.58_{\pm1.14}$ | $93.20_{\pm1.01}$ | $72.78_{\pm2.56}$ | $54.01_{\pm1.15}$ | $76.98_{\pm0.82}$ | $40.01_{\pm1.95}$ | $60.17_{\pm1.38}$ |

> **Observation 1 (Undirected Graphs)** *As shown in Table 2, under the unified encoder–decoder training framework, network embedding methods exhibit strong competitiveness on undirected link prediction tasks, requiring only minor hyperparameter adjustments. In many cases, they even surpass state-of-the-art GNN models.*

Prior work suggests that network embedding methods perform significantly worse than GNN approaches on undirected link prediction tasks. However, our re-implementations of classical embedding methods reach top-2 performance on all nine benchmark datasets, surpassing GNNs on eight and achieving state-of-the-art results. Specifically, node2vec* outperforms HL-GNN (Zhang et al., 2024) and achieves the best performance on Cora, CiteSeer and ogbl-collab, with accuracy improvements of 26.07%, 26.55%, and 20.04%, respectively. Similarly, DeepWalk* also shows notable improvements, with accuracy gains ranging from 0.27% to 52.59%. Although LINE* was not included in previous baselines, our results show that it is a surprisingly strong competitor, achieving the best performance on Computers, ogbl-ddi, ogbl-ppa, and ogbl-citation2.

> **Observation 2 (Directed Graphs)** *As shown in Table 3, network embedding methods, particularly those based on random walks, are naturally suited to directed graphs because the walk process inherently respects edge direction. In contrast to GNNs, which often require specialized mechanisms for directional message passing, these methods can be applied directly without modification and often achieve better performance than GNNs designed for directed link prediction.*

Across the seven directed graph datasets, our re-implementations of three classical embedding methods achieve first-place performance on five. Specifically, on the Photo-D, WikiCS, and Epinions datasets, node2vec* not only outperforms all GNN methods but also exceeds STRAP (Yin & Wei, 2019), another embedding-based approach, by 3.62%, 0.71%, and 1.18% in accuracy, respectively. While prior work (He et al., 2025) has shown that some embedding methods can perform well on directed link prediction tasks, our results further reinforce this conclusion. Furthermore, node2vec* achieves the highest performance among all baselines on five datasets: Cora-ML, Photo-D, Computers-D, WikiCS, and Epinions, demonstrating consistent superiority across diverse graph types. Both LINE* and DeepWalk* also demonstrate strong competitiveness. Notably, LINE* achieves 42.24% and 59.89% accuracy on Slashdot and Epinions, respectively, substantially outperforming most GNN models and ranking second. Similarly, DeepWalk* ranks second on Cora-ML with 92.42% accuracy, just behind node2vec* (92.58%), and significantly ahead of traditional and directed GNN baselines.

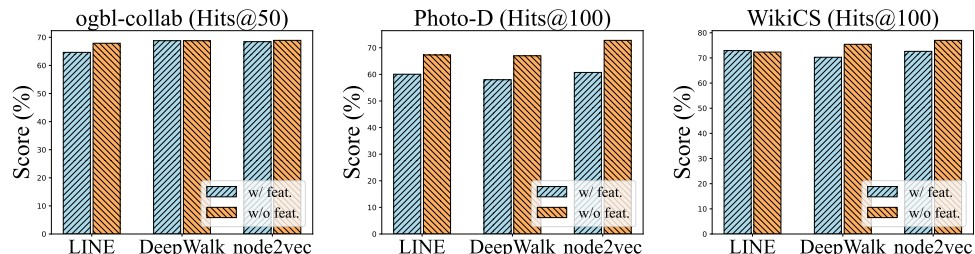

Figure 1: Link prediction performance comparison with and without incorporating node features.

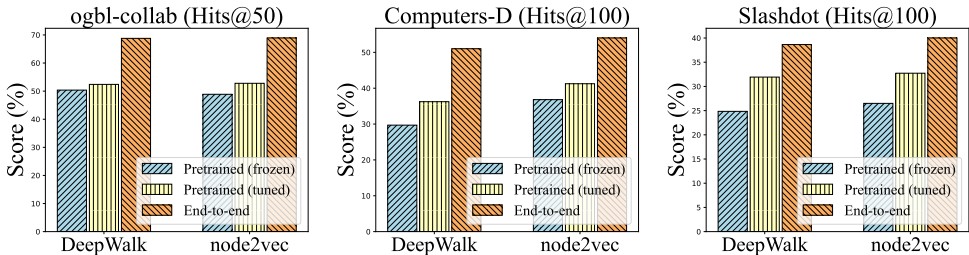

Figure 2: Performance comparison of three training strategies for network embedding methods: Pretrained with frozen embeddings, pretrained with fine-tuning, and fully end-to-end training.

**Observation 3** *As shown in Figure 1, the contribution of node features to link prediction is limited and inconsistent. In many cases, incorporating node features yields no improvement or even degrades performance, suggesting that structure alone is often sufficient for link prediction.*

We study the impact of node features on link prediction performance by comparing models trained with and without attribute information. For LINE*, DeepWalk* and node2vec*, we add node features by concatenating each node's feature vector with its learned embedding, then feeding the result into the decoder for link scoring, as described in Section 3. As shown in Figure 1, the inclusion of node features often leads to marginal or even negative changes in performance. In several cases, such as ogbl-collab and WikiCS, adding node features bring no significant gain and can even reduce accuracy. One possible explanation is that link prediction is typically framed as a binary classification task focused on predicting the existence of links between node pairs. When the graph structure already provides strong topological signals, extra features may add little value, especially if sparse, noisy, or misaligned with the link structure. These findings align with prior observations that the utility of node features in link prediction is often limited or task-specific (He et al., 2025; Zhu et al., 2024b). Importantly, in scenarios where node features provide little benefit, classical embedding methods remain surprising effective due to their strong ability to capture structure. These results suggest that such methods deserve renewed attention as robust solutions for link prediction.

## 5.2 INFLUENCE OF TRAINING STRATEGY ON LINK PREDICTION PERFORMANCE

**Observation 1: End-to-end training consistently improves the performance of network**. In prior work, network embedding methods such as DeepWalk and node2vec are often evaluated using a pretrained-frozen setup, where node embeddings are first learned through unsupervised pretraining and then used as fixed input features for a downstream classifier (e.g., MLP), with the embedding table kept frozen during training. This evaluation protocol differs markedly from the training of modern GNNs (Zhang et al., 2024; Kipf & Welling, 2016; Veličković et al., 2018), which are trained end-to-end. It is also worth noting that the original objective of DeepWalk (Perozzi et al., 2014) and node2vec (Grover & Leskovec, 2016) is to model node proximity, which directly aligns with link prediction. Freezing the encoder during downstream training breaks this alignment by decoupling the encoder from the task it was designed to support. We argue that network embedding methods should be trained end-to-end, or at least fine-tuned during downstream optimization, to fully leverage their capacity within a modern encoder–decoder framework, as described in Section 3. As shown in Figure 2, the Pretrained (tuned) strategy, where pretrained embeddings are updated during training, yields significantly better performance than the Pretrained (frozen) variant. Nevertheless, the highest performance is consistently achieved with fully End-to-end training, where both the encoder and decoder are jointly optimized for the task.

Table 4: Comparison of decoding methods for link prediction. "cat" denotes vector concatenation and $\odot$ denotes the Hadamard product. Best result per model group is highlighted in **bold**.

| Method | Decoder | CiteSeer Hits@100 | Computers AUC | Photo-D Hits@100 | Slashdot Hits@100 |
|---|---|---|---|---|---|
| LINE* | DOT | $84.40_{\pm 0.75}$ | $98.18_{\pm 0.02}$ | $66.27_{\pm 2.44}$ | $41.69_{\pm 0.92}$ |
| | MLP (cat) | $67.58_{\pm 3.40}$ | $98.74_{\pm 0.02}$ | $10.51_{\pm 2.83}$ | $33.20_{\pm 1.02}$ |
| | MLP ($\odot$) | $\mathbf{95.71_{\pm 0.94}}$ | $\mathbf{98.97_{\pm 0.01}}$ | $\mathbf{67.28_{\pm 2.59}}$ | $\mathbf{42.24_{\pm 0.64}}$ |
| DeepWalk* | DOT | $90.90_{\pm 1.45}$ | $98.60_{\pm 0.02}$ | $\mathbf{71.56_{\pm 1.96}}$ | $35.03_{\pm 0.90}$ |
| | MLP (cat) | $78.13_{\pm 3.57}$ | $98.70_{\pm 0.01}$ | $13.28_{\pm 1.21}$ | $31.83_{\pm 1.05}$ |
| | MLP ($\odot$) | $\mathbf{95.25_{\pm 1.91}}$ | $\mathbf{98.82_{\pm 0.01}}$ | $71.14_{\pm 1.61}$ | $\mathbf{38.65_{\pm 0.97}}$ |
| node2vec* | DOT | $90.70_{\pm 1.34}$ | $98.57_{\pm 0.02}$ | $\mathbf{72.78_{\pm 2.56}}$ | $35.16_{\pm 0.79}$ |
| | MLP (cat) | $78.20_{\pm 3.22}$ | $98.73_{\pm 0.01}$ | $13.07_{\pm 1.48}$ | $32.72_{\pm 1.32}$ |
| | MLP ($\odot$) | $\mathbf{95.89_{\pm 1.32}}$ | $\mathbf{98.83_{\pm 0.02}}$ | $72.69_{\pm 1.07}$ | $\mathbf{40.01_{\pm 1.95}}$ |

**Observation 2: A more expressive decoder, such as an MLP, often leads to improved performance**. Traditional embedding methods typically rely on a simple dot product to compute link prediction scores. However, as discussed in Section 3, the decoder is not the primary factor distinguishing different embedding methods. Within the modern encoder-decoder framework, embedding methods should be understood as encoders, and may incorporate task-specific data augmentation strategies. The decoder, by contrast, remains modular and can be selected based on dataset characteristics and task requirements. In Table 4, we compare three decoder designs: DOT (dot product), MLP (concat), and MLP (Hadamard product). The results show that MLP with Hadamard product consistently achieves the best performance on most datasets, highlighting the importance of using a more expressive decoder to fully leverage the learned embeddings. Although MLPs are powerful in modeling non-linear relationships, the MLP (concat) decoder relies on the model to implicitly learn interactions between node pairs from concatenated embeddings, which is often less effective than explicitly modeling pairwise interactions via the Hadamard product.

**Observation 3: Pairwise ranking losses typically lead to improved performance on ranking-based metrics such as Hit Rate**. The choice of loss function plays a crucial role in link prediction, as it directly affects the quality of the learned embeddings. In Figure 3, we evaluate the performance of LINE*, DeepWalk*, and node2vec* on the Computers-D dataset, comparing two types of loss functions: a pointwise loss (BCE) and pairwise ranking losses (BPR and AUC loss). The results show that BPR and AUC, which explicitly model the relative ranking between positive and negative samples, generally achieve better results when evaluated using ranking-oriented metrics such as Hits@100. In contrast, BCE focuses on minimizing the discrepancy between predicted probabilities and binary labels on a per-sample

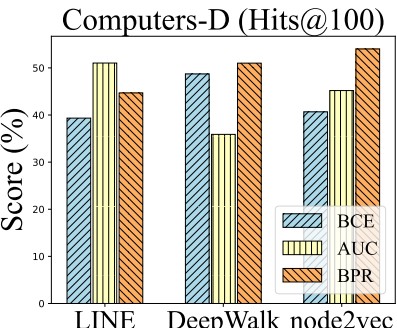

Figure 3: Comparison of loss functions for link prediction on Computers-D.

basis, without directly optimizing for the ranking of positive instances. Due to space constraints, additional experimental results and detailed comparisons are provided in the Appendix D.5.

## 6 CONCLUSION

In this work, we revisit classical network embedding methods such as LINE, DeepWalk, and node2vec in link prediction tasks. Contrary to the widely held belief that GNN-based models significantly outperform embedding-based approaches, our empirical results demonstrate that, under a unified encoder-decoder training framework with appropriate loss functions and decoders, network embedding methods can achieve competitive, and in many cases, state-of-the-art performance on both undirected and directed graphs. We further analyze the impact of training strategies, decoder architectures, and loss functions, revealing that end-to-end training, expressive decoders, and task-aligned ranking losses play a critical role in optimizing performance. Our findings not only call for more rigorous comparisons between classical and modern graph learning methods but also highlight the enduring value of embedding-based approaches when they are properly trained.

## REPRODUCIBILITY STATEMENT

Detailed descriptions of the experimental setup, implementation procedures, and hyper-parameter configurations are provided in Section 4 and Appendix D. To support repro-ducibility, we release an anonymous code repository at `https://www.dropbox.com/scl/fo/ulxyvt9kovb2ll2y919f2/AEFIfea4VvpHAgZGrwkFcTE?rlkey=xh5r14as78kbec5prv1rdjl8s&st=npb3sq4a&dl=0`.

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

## A   USAGE OF LLMS

In this work, we use LLMs as grammar checkers for article writing and polishing. LLMs are not used for idea discovery or direct content generation.

## B   NOTATION

We summarize the primary notations used throughout the paper in Table 5.

Table 5: Summary of notations used in this paper.

| Symbol | Description |
|--------|-------------|
| $G = (V, E)$ | Input graph with node set $V$ and edge set $E$ |
| $n = |V|$ | Number of nodes |
| $m = |E|$ | Number of edges |
| $\mathbf{A} \in \mathbb{R}^{n \times n}$ | Adjacency matrix of the graph |
| $\mathbf{X} \in \mathbb{R}^{n \times h}$ | Input feature matrix with $h$-dimensional features for $n$ nodes |
| $\mathbf{E} \in \mathbb{R}^{n \times d}$ | Learned embedding matrix; $d$ is the embedding size |
| $f$ | Encoder function (e.g., GNN or embedding lookup) |
| $g$ | Decoder function for computing link scores |
| $\hat{y}_{ij}$ | Predicted link score between node $i$ and node $j$ |
| $y_{ij} \in \{0, 1\}$ | Ground-truth label for the link between node $i$ and node $j$ |
| $\odot$ | Element-wise (Hadamard) product |
| $[\cdot \| \cdot]$ | Concatenation operation |
| $\sigma$ | Sigmoid function |

## C   COMPLEXITY ANALYSIS

We present a detailed analysis of the time and space complexity of our proposed method in comparison with representative baselines. Table 6 summarizes the theoretical complexities, where $n$ denotes the number of nodes, $m$ the number of edges, $d$ the embedding dimension, $L$ the number of MLP or GCN layers, $F$ the polynomial order, $r$ the number of walks per node, and $K$ the walk length.

Table 6: Comparison of Time and Space Complexity.

|  | Time Complexity | Space Complexity |
|--------|-----------------|------------------|
| GCN | $O(Lmd + Lnd^2)$ | $O(nd + m)$ |
| HL-GNN | $O(Lmd + Lnd^2)$ | $O(nd + m)$ |
| SDGAE | $O(Fmd + Lnd^2)$ | $O(nd + m)$ |
| LINE* | $O(Lnd^2)$ | $O(nd)$ |
| DeepWalk* | $O(nrK + Lnd^2)$ | $O(nd)$ |
| node2vec* | $O(nrK + Lnd^2)$ | $O(nd)$ |

Our method revisits and unifies classical embedding approaches such as LINE, DeepWalk, and node2vec within a unified framework, followed by a lightweight MLP for downstream tasks. In contrast to GCN-based methods, these approaches circumvent message passing and graph convolution, thereby obviating the need to compute or store multi-hop neighborhoods. This design results in a more streamlined and scalable architecture, particularly advantageous for large-scale graphs.

For GCN-based methods, the space complexity is $O(nd + m)$, where $O(nd)$ accounts for storing node representations and $O(m)$ corresponds to maintaining the sparse adjacency matrix required for message passing. In contrast, the embedding-based approach eliminates graph convolution and neighborhood aggregation, thereby removing the need of storing the adjacency matrix during training. Consequently, the space complexity reduces to $O(nd)$, which is dominated by the node embedding table.

## D  EXPERIMENT DETAILS

### D.1  CODE AVAILABILITY

To ensure fair comparison, we implement all models within standardized training and evaluation frameworks. Specifically, for the Planetoid and Amazon datasets, we base our implementations of LINE*, DeepWalk*, and node2vec* on the official HL-GNN codebase [1]. For OGB datasets, we follow the code structure used in the OGB leaderboard [2]. For directed graphs, our implementations are based on the DirLinkBench framework [3]. All code and configurations used in our experiments are available at `https://www.dropbox.com/scl/fo/ulxyvt9kovb2ll2y919f2/AEFIfea4VvpHAgZGrwkFcTE?rlkey=xh5r14as78kbec5prv1rdjl8s&st=npb3sq4a&dl=0`.

### D.2  BASELINE REUSE AND EXPERIMENTAL CONSISTENCY

It is both reasonable and widely accepted to reuse baseline results reported in prior studies, particularly when comparable settings are adopted, as in the case of HL-GNN. This practice enhances reproducibility and facilitates more efficient and credible progress in the field (Wang & Zhang, 2022; Wang et al., 2024). To further ensure fairness and consistency, we are rerunning all methods under a unified experimental setting. Due to time constraints, only a subset of experiments has been completed so far. As shown in Table 7, the reproduced results are broadly consistent with those reported in the original publications.

Table 7: Comparison of reproduced and reported results.

| Dataset Metric | Cora Hits@100 | Citeseer Hits@100 | Photo AUC | Computer AUC | ogbl-collab Hits@50 |
|---|---|---|---|---|---|
| BUDDY (reported) | $88.00 \pm 0.44$ | $92.93 \pm 0.27$ | $99.05 \pm 0.21$ | $98.69 \pm 0.34$ | $65.94 \pm 0.58$ |
| BUDDY | $85.69 \pm 0.43$ | $91.60 \pm 1.38$ | $89.33 \pm 1.60$ | $98.41 \pm 0.44$ | $66.01 \pm 0.46$ |
| HL-GNN (reported) | $94.22 \pm 1.64$ | $94.31 \pm 1.51$ | $99.11 \pm 0.07$ | $98.82 \pm 0.21$ | $68.11 \pm 0.54$ |
| HL-GNN | $93.84 \pm 1.36$ | $93.80 \pm 1.74$ | $98.93 \pm 0.09$ | $98.78 \pm 0.23$ | $68.23 \pm 0.61$ |

### D.3  IMPLEMENTATION DETAILS

To ensure fair comparison and reproducibility, we follow the experimental setups of HL-GNN (Zhang et al., 2024) for undirected graphs and DirLinkBench (He et al., 2025) for directed graphs. Our re-implementations of LINE*, DeepWalk*, and node2vec* are integrated into the unified encoder–decoder framework described in Section 3 and trained in an **end-to-end** manner, where both the embedding matrix and decoder parameters are jointly optimized for the link prediction objective. Algorithm 1 details the full training procedure for node2vec* using an MLP decoder and binary cross-entropy loss. LINE* and DeepWalk* follow analogous procedures. The process begins with randomly initialized trainable embeddings (line 2), which are optionally concatenated with node features, if available (lines 3–7). Additional training edges are then generated via biased random walks (lines 9–11), following the original node2vec sampling strategy. These walks are converted into training samples using a Skip-Gram-style procedure, resulting in an augmented edge set $E_{\text{aug}}$. In the final stage (lines 13–19), supervised link prediction is performed by training a MLP decoder on both the original and augmented edges.

To illustrate the differences between our end-to-end training approach and the commonly used evaluation protocol for network embedding methods, Algorithm 2 outlines the standard training procedure traditionally applied to node2vec (Grover & Leskovec, 2016) in link prediction benchmarks. In this setup, the embedding matrix $\mathbf{E}$ is first trained independently using the Skip-Gram objective with negative sampling, which encourages similar embeddings for nodes that co-occur in random

---

[1] `https://github.com/LARS-research/HL-GNN`

[2] `https://ogb.stanford.edu/docs/leader_linkprop`

[3] `https://github.com/ivam-he/DirLinkBench-SDGAE`

walk sequences. This initial embedding phase is shown in lines 2–9. Afterward, the resulting embedding matrix is frozen, and a separate MLP decoder is trained for the link prediction task using the **fixed** embeddings (lines 19-26). Crucially, this decoupled setup breaks the alignment between the embedding objective and the downstream task. Since the Skip-Gram objective already captures pairwise node interactions in a manner closely related to link prediction, freezing the embeddings may underutilize their task-relevant potential and lead to suboptimal performance.

To further bridge the gap between classical network embedding methods and modern training paradigms, we explore a fine-tuning variant of node2vec, as described in Algorithm 3. In this setting, the embedding matrix is initialized with pretrained node2vec embeddings (line 3), but unlike the frozen protocol, the embeddings are **updated** during downstream training. This approach retains the structural inductive bias of node2vec while enabling task-specific adaptation. As shown in Figure 2, fine-tuning consistently outperforms the frozen variant and narrows the performance gap with fully end-to-end models.

---

**Algorithm 1:** node2vec* (end-to-end) for Link Prediction with MLP

---

**Input:** Graph $G = (V, E)$, base node features $\mathbf{X}$, optional pretrained embedding $\mathbf{E}_{\text{pre}}$, MLP
      hidden size $h$, dropout $\delta$, epochs $T$

**Output:** Trained embeddings $\mathbf{E}_\theta$ and MLP parameters $\theta$

1  // Model Initialization;

2  Initialize trainable embedding $\mathbf{E}_\theta \in \mathbb{R}^{|V| \times d}$;

3  **if** *original features* $\mathbf{X}$ *exist* **then**

4    |  $\mathbf{X}' \leftarrow [\mathbf{X} \,\|\, \mathbf{E}_\theta]$;

5  **end**

6  **else**

7    |  $\mathbf{X}' \leftarrow \mathbf{E}_\theta$;

8  **end**

9  // Data Augmentation using node2vec;

10  For each $v_i \in V$, sample walk sequences $\mathcal{W}_{v_i}$ with length $K$;

11  Generate $E_{\text{aug}}$ by additional training pairs $(u, v)$ from $\mathcal{W}_{v_i}$ (skip-gram style);

12  Merge $E_{\text{aug}}$ into training edge set;

13  // End-to-End Training;

14  Initialize MLP predictor $g_\theta$ with $L$ layers, input dim = dim($\mathbf{X}'$), hidden dim $h$, dropout $\delta$;

15  **for** *epoch* $\leftarrow 1$ **to** $T$ **do**

16    Sample positive and negative edge batches $(u, v)$ and $(u', v')$;

17    Get embeddings: $x_u \leftarrow \mathbf{X}'[u]$, $x_v \leftarrow \mathbf{X}'[v]$;

18    Compute predictions: $\hat{y}_{uv} \leftarrow g_\theta(x_u \cdot x_v)$;

19    Compute loss: $\mathcal{L} = -\log \hat{y}_{uv} - \log(1 - \hat{y}_{u'v'})$;

20    Update all parameters $(\theta, \mathbf{E}_\theta)$ jointly via gradient descent;

21  **end**

---

---

**Algorithm 2:** node2vec (fixed) for Link Prediction with MLP

---

**Input:** Graph $G = (V, E)$, walk length $K$, embedding size $d$, walks per vertex $\gamma$, return parameter $p$, in-out parameter $q$, window size $w$
MLP hidden size $h$, number of layers $L$, dropout $\delta$, learning rate $\eta$, training epochs $T$
**Output:** Trained MLP parameters $\theta$

1   // node2vec Embedding Learning;
2   Sample embedding matrix $\mathbf{E} \sim \mathcal{U}^{n \times d}$;
3   **for** $i \leftarrow 1$ **to** $\gamma$ **do**
4     $\mathcal{O} \leftarrow \texttt{Shuffle}(V)$;
5     **foreach** $v_i \in \mathcal{O}$ **do**
6       $\texttt{RW}_{v_i} \leftarrow \texttt{BiasedRandomWalk}(G, v_i, K, p, q)$;
7       $\texttt{SkipGram}(\mathbf{E}, \texttt{RW}_{v_i}, w)$;
8     **end**
9   **end**
10   Save $\mathbf{E}$ to `embedding.pt`;

11   // Link Prediction with MLP;
12   Load $\mathbf{E}$ from `embedding.pt`;
13   **if** *original features* $\mathbf{X}$ *exist* **then**
14     $\mathbf{X'} \leftarrow [\mathbf{X} \,\|\, \mathbf{E}]$;
15   **end**
16   **else**
17     $\mathbf{X'} \leftarrow \mathbf{E}$;
18   **end**
19   Initialize MLP predictor $g_\theta$ with $L$ layers, input dim = dim($\mathbf{X'}$), hidden dim $h$, dropout $\delta$;
20   **for** *epoch* $\leftarrow 1$ **to** $T$ **do**
21     Sample positive and negative edge batches $(u, v)$ and $(u', v')$;
22     Get embeddings: $x_u \leftarrow \mathbf{X'}[u]$, $x_v \leftarrow \mathbf{X'}[v]$;
23     Compute predictions: $\hat{y}_{uv} \leftarrow g_\theta(x_u \cdot x_v)$;
24     Compute loss: $\mathcal{L} = -\log \hat{y}_{uv} - \log(1 - \hat{y}_{u'v'})$;
25     Update MLP parameters $\theta$ via gradient descent;
26   **end**

---

**Algorithm 3:** node2vec (tuned) for Link Prediction with MLP

---

**Input:** Graph $G = (V, E)$, hidden size $h$, layers $L$, dropout $\delta$, learning rate $\eta$, training epochs $T$, edge splits
**Output:** Trained MLP parameters $\theta$, fine-tuned embedding $\mathbf{E}_\theta$

1   // Fine-tunable Embedding Initialization;
2   Load $\mathbf{E}_{\text{pre}}$ from `embedding.pt` generated via node2vec (omitted here);
3   $\mathbf{E}_\theta \leftarrow \texttt{InitializeFrom}(\mathbf{E}_{\text{pre}}, \texttt{trainable=True})$;
4   **if** *original features* $\mathbf{X}$ *exist* **then**
5     $\mathbf{X'} \leftarrow [\mathbf{X} \,\|\, \mathbf{E}_\theta]$;
6   **end**
7   **else**
8     $\mathbf{X'} \leftarrow \mathbf{E}_\theta$;
9   **end**

10   // Link Prediction with MLP;
11   Initialize MLP predictor $g_\theta$ with $L$ layers, input dim = dim($\mathbf{X'}$), hidden dim $h$, dropout $\delta$;
12   **for** *epoch* $\leftarrow 1$ **to** $T$ **do**
13     Sample positive and negative edge batches $(u, v)$ and $(u', v')$;
14     Get embeddings: $x_u \leftarrow \mathbf{X'}[u]$, $x_v \leftarrow \mathbf{X'}[v]$;
15     Compute predictions: $\hat{y}_{uv} \leftarrow g_\theta(x_u \cdot x_v)$;
16     Compute loss: $\mathcal{L} = -\log \hat{y}_{uv} - \log(1 - \hat{y}_{u'v'})$;
17     Update all parameters $(\theta, \mathbf{E}_\theta)$ jointly via gradient descent;
18   **end**

---

## D.4 HYPERPARAMETERS

We report the hyperparameter configurations used for each model in Tables 8 and 9, corresponding to the undirected and directed link prediction tasks, respectively. All values are selected based on validation performance following the settings of HL-GNN (Zhang et al., 2024) and DirLinkBench (He et al., 2025). For most models, we adopt hyperparameters from prior benchmark implementations to ensure consistency.

Table 8: Hyperparameter settings for all models on undirected link prediction datasets.

| Dataset | Method | Walk length $K$ | $p$ | $q$ | Decoder | Layer $L$ | Hidden dim | Dropout rate | LR | Loss function |
|---|---|---|---|---|---|---|---|---|---|---|
| Cora | LINE | - | - | - | MLP ($\odot$) | 5 | 4096 | 0.4 | 0.001 | BCE |
| | DeepWalk | 3 | 1 | 1 | MLP ($\odot$) | 5 | 4096 | 0.4 | 0.001 | BCE |
| | node2vec | 3 | 1 | 2 | MLP ($\odot$) | 5 | 4096 | 0.4 | 0.001 | BCE |
| CiteSeer | LINE | - | - | - | MLP ($\odot$) | 3 | 4096 | 0.5 | 0.005 | BCE |
| | DeepWalk | 3 | 1 | 1 | MLP ($\odot$) | 3 | 4096 | 0.5 | 0.005 | BCE |
| | node2vec | 3 | 1 | 2 | MLP ($\odot$) | 3 | 4096 | 0.5 | 0.005 | BCE |
| Pubmed | LINE | - | - | - | MLP ($\odot$) | 4 | 1024 | 0.1 | 0.008 | BCE |
| | DeepWalk | 3 | 1 | 1 | MLP ($\odot$) | 4 | 1024 | 0.1 | 0.008 | BCE |
| | node2vec | 3 | 1 | 2 | MLP ($\odot$) | 4 | 1024 | 0.1 | 0.008 | BCE |
| Photo | LINE | - | - | - | MLP ($\odot$) | 4 | 512 | 0.4 | 0.0008 | BCE |
| | DeepWalk | 3 | 1 | 1 | MLP ($\odot$) | 5 | 512 | 0.3 | 0.001 | BCE |
| | node2vec | 3 | 1 | 2 | MLP ($\odot$) | 5 | 512 | 0.3 | 0.001 | BCE |
| Computers | LINE | - | - | - | MLP ($\odot$) | 5 | 512 | 0.4 | 0.0008 | BCE |
| | DeepWalk | 3 | 1 | 1 | MLP ($\odot$) | 5 | 512 | 0.4 | 0.0008 | BCE |
| | node2vec | 3 | 1 | 2 | MLP ($\odot$) | 5 | 512 | 0.4 | 0.0008 | BCE |
| ogbl-collab | LINE | - | - | - | DOT | - | - | - | 0.0001 | AUC |
| | DeepWalk | 3 | 1 | 1 | DOT | - | - | - | 0.0001 | AUC |
| | node2vec | 3 | 1 | 2 | DOT | - | - | - | 0.0001 | AUC |
| ogbl-ddi | LINE | - | - | - | MLP ($\odot$) | 4 | 512 | 0.2 | 0.001 | AUC |
| | DeepWalk | 3 | 1 | 1 | MLP ($\odot$) | 4 | 512 | 0.2 | 0.001 | AUC |
| | node2vec | 3 | 1 | 5 | MLP ($\odot$) | 4 | 512 | 0.2 | 0.001 | AUC |
| ogbl-ppa | LINE | - | - | - | MLP ($\odot$) | 3 | 512 | 0.5 | 0.001 | AUC |
| | DeepWalk | 3 | 1 | 1 | MLP ($\odot$) | 3 | 512 | 0.5 | 0.001 | AUC |
| | node2vec | 3 | 1 | 5 | MLP ($\odot$) | 3 | 512 | 0.5 | 0.001 | AUC |
| ogbl-citation2 | LINE | - | - | - | MLP ($\odot$) | 3 | 256 | 0.5 | 0.001 | AUC |
| | DeepWalk | 3 | 1 | 1 | MLP ($\odot$) | 3 | 256 | 0.5 | 0.001 | AUC |
| | node2vec | 3 | 1 | 5 | MLP ($\odot$) | 3 | 256 | 0.5 | 0.001 | AUC |

Table 9: Hyperparameter settings for all models on directed link prediction datasets.

| Dataset | Method | Walk length $K$ | $p$ | $q$ | Decoder | Layer $L$ | Hidden dim | Dropout rate | LR | Loss function |
|---|---|---|---|---|---|---|---|---|---|---|
| Cora-ML | LINE | - | - | - | DOT | - | - | - | 0.005 | BPR |
| | DeepWalk | 3 | 1 | 1 | DOT | - | - | - | 0.01 | BPR |
| | node2vec | 3 | 1 | 5 | DOT | - | - | - | 0.01 | BPR |
| CiteSeer-D | LINE | - | - | - | DOT | - | - | - | 0.005 | BPR |
| | DeepWalk | 3 | 1 | 1 | DOT | - | - | - | 0.005 | BPR |
| | node2vec | 3 | 1 | 5 | DOT | - | - | - | 0.005 | BPR |
| Photo-D | LINE | - | - | - | MLP ($\odot$) | 2 | 64 | 0.5 | 0.005 | AUC |
| | DeepWalk | 3 | 1 | 1 | MLP ($\odot$) | 2 | 64 | 0.5 | 0.005 | BPR |
| | node2vec | 3 | 1 | 5 | MLP ($\odot$) | 2 | 64 | 0.5 | 0.005 | BPR |
| Computers-D | LINE | - | - | - | DOT | - | - | - | 0.005 | AUC |
| | DeepWalk | 3 | 1 | 1 | MLP ($\odot$) | 1 | - | - | 0.005 | BPR |
| | node2vec | 3 | 1 | 5 | MLP ($\odot$) | 1 | - | - | 0.005 | BPR |
| WikiCS | LINE | - | - | - | MLP ($\odot$) | 2 | 64 | 0.5 | 0.005 | AUC |
| | DeepWalk | 3 | 1 | 1 | DOT | | | | 0.005 | BPR |
| | node2vec | 3 | 1 | 5 | DOT | | | | 0.005 | BPR |
| Slashdot | LINE | - | - | - | MLP ($\odot$) | 1 | - | - | 0.005 | BPR |
| | DeepWalk | 3 | 1 | 1 | MLP ($\odot$) | 1 | - | - | 0.005 | BPR |
| | node2vec | 3 | 1 | 5 | MLP ($\odot$) | 1 | - | - | 0.005 | BPR |
| Epinions | LINE | - | - | - | DOT | - | - | - | 0.005 | BCE |
| | DeepWalk | 3 | 1 | 1 | DOT | - | - | - | 0.005 | BCE |
| | node2vec | 3 | 1 | 5 | DOT | - | - | - | 0.005 | BCE |

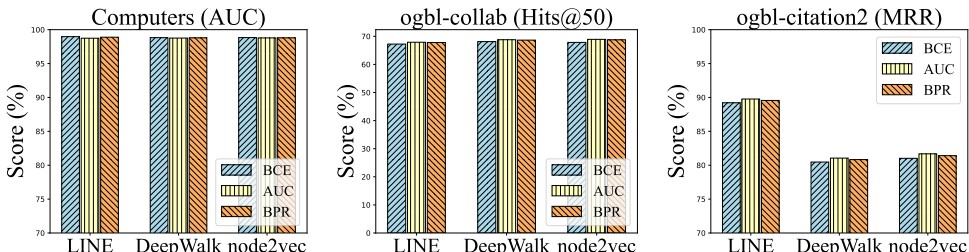

Figure 4: Comparison of loss functions for link prediction.

## D.5  EXTENDED ANALYSIS: LOSS FUNCTION COMPARISON

To complement Observation 3 in Section 5.2, we conduct additional experiments to compare the effects of three loss functions: BCE, BPR, and the AUC loss. We evaluate their impact on three representative datasets: Computers-D, ogbl-collab, and ogbl-citation2, the results are shown in Figure 4.

When AUC is used as the evaluation metric, the choice of loss function has relatively limited impact, with BCE performing competitively in most cases. In contrast, for ranking-based metrics such as Hits@50 or MRR, pairwise losses like BPR and AUC tend to outperform BCE in most scenarios. This supports the intuition that pairwise losses are better aligned with ranking objectives, as they explicitly optimize the relative ordering between positive and negative samples. These findings underscore the importance of aligning the loss function with the evaluation metric—particularly for link prediction tasks where ranking performance is critical.

