# OpenReview forum: "Network Embedding Methods are Strong Baselines for Link Prediction"
_ICLR.cc/2026/Conference — ICLR 2026 Conference Withdrawn Submission_

### Official Review · Reviewer_2qZU · 2025-10-25

**Soundness:** 3
**Presentation:** 4
**Contribution:** 2
**Rating:** 4
**Confidence:** 3

**Summary:**

This paper shows that, within a unified encoder–decoder training protocol, classic network embeddings (LINE/DeepWalk/node2vec) can outperform recent GNNs on most link-prediction benchmarks

**Strengths:**

Good experiments;
Replicable;
They paper did very comprehensive comparision on dataset and methods;

**Weaknesses:**

However, I do not think that testing different methods and datasets, by itself, makes a significant contribution to society.
First, the novelty is limited: the paper reads more like testing and survey work rather than a new theory. These experimental results are certainly useful, but the paper still need to demonstrate the significance of the contribution. The current version largely repeats existing methods, so it feels more like a knowledge checklist (can't find anything new from line 77 to line 248) with comprehensive check that things work, rather than a claim that something new has been discovered, explained, and then validated with further experiments. These style of surveying could be useful if the paper is still the first one to do this (I haven’t checked whether similar surveys exist, but it’s very likely they do.)

Regarding the observations:

Observation 1: It is anticipated that supervised representation learning would outperform unsupervised learning.

Observation 2: The paper itself also indicates that this result was anticipated.

Observation 3: This could be a very significant finding if it can be proved to hold in general, however, the explanation is weak. Sometimes node features help and sometimes structure helps, so the current result may just reflect one particular case in the test suite. I checked the analysis and appendix, but the paper focuses on the loss function rather than this observation. At a minimum, the paper should consider how to evaluate when structure is more informative than node features.

**Questions:**

The work appears complete for its scope and stated aims; I have no specific questions except for the above comments.

---

### Official Review · Reviewer_EMLW · 2025-10-28

**Soundness:** 2
**Presentation:** 3
**Contribution:** 2
**Rating:** 4
**Confidence:** 3

**Summary:**

This paper reevaluates the performance of several classic embedding-based models for link prediction, namely LINE, DeepWalk, and node2vec. It proposes a unified framework for the training process and conducts an empirical evaluation of these three models across 16 benchmark datasets. By carefully tuning hyperparameters under different settings, the paper shows that these baseline embedding models can outperform GNN-based models in most cases. In addition, it investigates the role of node features and different training strategies for node embeddings, yielding some interesting insights.

**Strengths:**

1. It is valuable that the paper revisits simple baseline methods, providing some new insights.


2. The experimental results are comprehensive, covering a wide range of datasets.


3. The unified training framework helps position these baselines in a broader methodological context.

**Weaknesses:**

1. The training details of the GNN models are not clearly explained. For example, do they follow the same configuration choices as the baselines, including decoders and loss functions?


2. It is also unclear whether the embeddings in the GNN models are treated as learnable parameters. For instance, are the embeddings trained end-to-end rather than being initialized from features? For embedding baseline models, we might expect improved performance due to the increased number of learnable parameters, which raises concerns about the fairness of the comparison between GNN and baseline models.


3. In the investigation of node features, the strategy of concatenating node features with the final embeddings seems questionable. How about an alternative design  to concatenate node features at the input stage and then use several MLP layers to map them into a new embedding space?

**Questions:**

please refer to the weaknesses

---

### Official Review · Reviewer_RsdF · 2025-10-31

**Soundness:** 2
**Presentation:** 3
**Contribution:** 2
**Rating:** 4
**Confidence:** 5

**Summary:**

This paper focuses on the problem of link prediction (LP). The main argument that the authors make, is that traditional network embedding (NE) models can actually outperform LP-Specific GNNs for LP. This is noteworthy, as it was previously assumed that such models struggled for this task. Specifically, the authors showed that by making multiple tweaks to network embedding models (via the loss, encoder, decoder, and training procedure), they can achieve competitive if not better performance on both undirected and directed link prediction. The authors show this for 3 popular embedding methods - Node2Vec, DeepWalk, and LINE. They further examine the effect that their various modifications have had on performance.

**Strengths:**

1. I think re-evaluating baseline methods is extremely important. Too often, we rely on old and outdated performance and simply assume that it is fine. However, through additional tuning and modifications, we can often achieve much higher performance of these baselines, calling into question whether any "real advances" were made in the design of recent methods. This is a long way of saying that I really like studies like this.

2. The performance is surprisingly good for both the undirected and directed datasets. One of the network embedding methods is often either the best (or nearly the best) for all datasets. This is really interesting.

3. The authors do a good job of showing the effect their various modifications have on performance. This includes end-to-end training, a MLP decoder, and the loss function. This better contextualizes why the performance of these models are able to improve relative to older reported performance.

**Weaknesses:**

1. A major problem I have, is that many of the modifications proposed in Section 3 are not unique to network embedding models and can be easily applied to other GNN methods. The problem here is to properly compare methods, we need them to be on equal footing. The authors pointed out several issues where network embedding models suffer, such as in using a dot-product encoder and not being trained in an end-to-end fashion. However, when we further make changes to the network embedding models that aren't found in the GNN models, we then need to further test the GNN methods with those changes for a fair comparison. This is analogous to making modifications to the embedding models that weren't in the original design. I.e., If we're training them end-to-end then why not also use BPR or AUC loss for GNN-based methods? There are a couple of examples:
            **(a)** As they noted the "skip-gram" style training essentially augments the training samples. However, this is something that can easily be applied when training GNNs. For example, (and please correct me if I'm wrong), but LINE does not use this strategy in their original implementation. However you use it in yours. So why not use it for the GNN methods?
            **(b)** A second difference is in the loss function. The loss function is nothing special and can be easily changed for any models. However, many LP-specific GNNs such as BUDDY, SEAL, Neo-GNN all use BCE. However, you experiments with {AUC, BCE, BPR}. Furthermore, per Figure 3 and 4, you show that it can significantly improve performance. As such, it should also be used for the other methods.

2. There is another major problem which isn't discussed. This could be included in the previous weakness but I figured it deserved it's own point. That is, network embedding methods have vastly more learnable parameters than other GNN methods. The reason is simple, embedding methods associate a $d$ dimensional embedding with each node while GNNs generally don't (especially the ones used in this study). Now this is tricky, because embedding are inherit to *embedding* methods. However, it still engenders an unfair comparison, as these methods will dwarf GNN methods in the number of parameters. For example, a GNN with $L$ layers and a max dimension of $d$ will have $O(L \cdot d^2)$ parameters, which since $L$ is quite small, $\approx O(d^2)$. On the other hand, the NE models have about $O(N \cdot d)$ parameters where $N$ is the number of nodes. For some datasets like Citation2 and PPA this can result in an extraordinary amount of parameters (I didn't see the dimension $d$ used in your experiments but for $d=50$, Citation2 would have about ~150 Million parameters). Having more parameters greatly enhances the modeling capabilities of any method. For example if you look at Table 4 in [1], they show that adding learnable embeddings can improve the NCN model by a non-trivial amount (note: I don't think they did any tuning so the performance can probably be improved more). This is all to say, the **vast disparity in learnable parameters between the methods do constitute an unfair comparison**.

3. Many relevant baselines are missing. This includes: NCN/NCNC [2], MPLP+ [3], LPFormer [4]. Of course, you can argue that the reported results are still not as good as what you report. However, again, it's not clear how they would compare under a fair comparison.

4. There are well known issues with the current evaluation setting used for undirected graphs (I'm unsure about the directed case). In [5], the authors introduce HeaRT which represents a more realistic and difficult evaluation setting. It has further gained traction and been used by many papers. While it's unclear if it would change the conclusions of the paper, it's nonetheless better to use a more appropriate and real-world setting for evaluating models.

5. This is a small weakness, but a downstream of the learnable embedding problem, is that the authors argue that structure alone is sufficient for LP. This is curious, because even using a basic MLP can often achieve respectable performance (see [5] for MLP performance on undirected graphs). This also contradicts evidence shown in [6], where in many cases feature similarity can indeed be needed for LP.


[1] "Reconsidering the Performance of GAE in Link Prediction." 2024.
[2] "Neural Common Neighbor with Completion for Link Prediction." ICLR, 2024.
[3] "Pure message passing can estimate common neighbor for link prediction." NeurIPS, 2024.
[4] "Lpformer: An adaptive graph transformer for link prediction." KDD, 2024.
[5] "Evaluating graph neural networks for link prediction: Current pitfalls and new benchmarking." NeurIPS, 2023.
[6] "Revisiting link prediction: a data perspective". ICLR, 2024.

**Questions:**

I know I'm asking a lot. As such, I've ordered my questions from most to least important

1. Can you try running some models using the same enhancements as the network embedding models + learnable node embeddings. Aagin, this is purely to get them on the same footing for a fair and equal comparison. This is the same as training the embedding models end-to-end which is not how they were originally designed. I know the authors include some results in Table 7, but they are clearly lacking both details and depth.

2. Try running your models on the HeaRT evaluation created in [1]. Again, as noted earlier, this evaluation is much more realistic and can give us a better idea of the real-world performance of the NE models used in the paper.

3. You should include the missing baselines (see weakness 3). Ideally, these would also be re-run using the same settings as the NE models.

4. I'm curious how much the performance increase is due to better negative sampling. Traditional methods (at least for undirected graphs), just randomly choose $L$ pairs of two nodes every epoch (see Figure 1 in [1] to see tha dangers of naive negative sampling). For example, LINE samples negatives, where it's biased to choosing less frequent nodes. I'm unsure how you are doing the negative sampling for the other NE methods. Nonetheless, I wonder if using this type of strategy for GNN methods would help.

[1] "Evaluating graph neural networks for link prediction: Current pitfalls and new benchmarking." NeurIPS, 2023.

---

### Official Review · Reviewer_7Dyj · 2025-11-01

**Soundness:** 4
**Presentation:** 3
**Contribution:** 2
**Rating:** 2
**Confidence:** 4

**Summary:**

The paper revisits classic network embedding algorithms—notably DeepWalk, node2vec, and LINE—and argues that when they are trained within a modern encoder–decoder framework using identical loss functions and decoders as GNN baselines, they can perform competitively on link-prediction tasks.

The authors benchmark these “traditional” embeddings on several directed and undirected datasets and claim that, under fair comparison, such embeddings outperform many graph neural networks.

**Strengths:**

1. **Comprehensive benchmarking**: The experiments cover many datasets and include widely used GNN baselines. The empirical setup is systematic and reproducible.

2. **Clear exposition of the encoder–decoder reinterpretation**: Framing embedding methods under a unified link-prediction pipeline makes the comparisons easy to follow.

**Weaknesses:**

1. **No novel theory or algorithm**: The paper introduces no new model, training objective, or mathematical analysis. Similar analysis were done in NeurIPS 2023 paper "Evaluating Graph Neural Networks for Link Prediction: Current Pitfalls and New Benchmarking."

2. **Missing related approaches**: Modern hybrid or proximity-based methods (e.g., PROXI: Challenging the GNNs for Link Prediction, TMLR 2025) and Autoencoders are not included in the discusions.

3. **Dataset bias (homophily)**:
The experimental results may primarily reflect the homophilic nature of the benchmark datasets used (e.g., Cora, Citeseer, Pubmed, OGB datasets). Classical embedding methods like DeepWalk and node2vec naturally perform well on homophilic graphs, where connected nodes tend to have similar features or labels. Therefore, their strong performance does not necessarily generalize to heterophilic or structurally diverse graphs, where message-passing GNNs typically excel.

4. **Paper Structure**: The paper does not contain "Related Work" section.

**Questions:**

1. Did you evaluate computational efficiency (training time, memory) versus GNNs to support your claims?
2. How does your method perform on heterophilic datasets (Texas, Cornell, Wisconsin, etc)?
3. What specific theoretical or methodological contribution distinguishes this work from existing benchmarking papers?

---

### Note · Authors · 2025-12-01

I have read and agree with the venue's withdrawal policy on behalf of myself and my co-authors.